# OpenReview forum: "Learning to Lie: Adversarial Attacks on Human-AI Teams and LLMs"
_ICLR.cc/2026/Conference — ICLR 2026 Poster_

### Official Review · Reviewer_gYP2 · 2025-10-16

**Soundness:** 3
**Presentation:** 3
**Contribution:** 2
**Rating:** 6
**Confidence:** 2

**Summary:**

The paper conduct a human study on the trust of humans to AIs, and how can AI potentially cheat human. The authors explores psychology-based and neural-network based belief dynamic modelling, and find both methods can effectively harm the human team. Finally, the authors replace human with LLMs and verifies its vulnerability to attacks.

**Strengths:**

The paper examines how can AI cheat human teams, an important and timely topic. The experiments are insightful and reveals important safety risks. The replacement of human to AIs and find advanced AI models cannot oversight cheating is a plus.

**Weaknesses:**

1. Clarity. While the experiments are solid, the takeaways and insights for practitioners are not immediately clear. Maybe summarization, highlight and mentioning potential application helps.

2. In the experiment, the authors claims that human will not trust AIs if it is "incorrect on an easy question". However, in subsequent analysis like Section 5.2, the strategic attack on whether the question is hard or not is not mentioned. How is the underlying process of deciding to lie or not? How is the lying answer given? If an wrong answer is given by AI in hard questions, maybe the trust by humans will decrease higher? I wonder if there are such analysis given.

3. Also in Section 5.2, the baseline compares cognitive model, MLP model and no attack. However, in the case without attack, the information provided by AI is a plus to the overall performance. So, I wonder if another baseline is needed for comparing with AI not providing any information at all.

4. Cognitive model seems uneffectivve in predicting the error comparing with equal weights baseline. Are there further experiments to verify the effectiveness of cognitive model? Current version seems highly ineffective.

I apologize that while I work on alignment and conduct experiments on human subjects, I am not familiar with existing works so please clarify if there is any mistake.

**Questions:**

See weaknesses.

---

> ### Author Response · Authors · 2025-11-22
> **Part 1: Response to Weaknesses 1,2**
>
> We thank the reviewer for the constructive assessment and for highlighting the importance of our topic and experimental findings. We appreciate the detailed comments on clarity, baseline choices, and the role of question difficulty, and we have reflected their suggestions and our responses below in our revised manuscript.
>
> **W1:** We appreciate the suggestion and will add a summary highlighting our contributions and potential applications. The key takeaway is that even relatively simple AI systems, when embedded in multi-human decision settings, can be used to harm decision-making processes. In real scenarios where complexity, scale, and access to user history are potentially much greater, these vulnerabilities may be amplified. Therefore, before deploying human-AI teams to safety critical domains, we must do more work to ensure the safety of human agents such as by devising more robust defense and monitoring strategies.
>
> **W2:** Our claim that the AI loses trust quickly when it is incorrect on an easy question is supported by qualitative and quantitative evidence as well as prior research. Qualitatively, participants expressed doubt in the AI when it missed easy questions. For example, after answering the first easy question wrong, one participant expressed “AI was wrong for the first question, so I don’t trust it fully.” Quantitatively, Appendix Table 1 shows the attacker primarily chose to lie on Medium/Hard questions, and additional analysis reveals that humans assigned almost 2x as many points to the AI on those problems. Together, these results indicate the AI strategically avoided lying on Easy questions due to predicting low long-term payoff. Finally, prior work has also found that trust in automation is more severely reduced when it errs on easy tasks [1].
>
> The attacker’s decision rule is straightforward. At each round, the planner estimates the cumulative future reward for two possible actions (i.e., tell the truth or lie) using DP rollouts and the internal trust model (either MLP or cognitive model). Then, the attacker selects the action with the higher expected cumulative reward. If the attacker lies, it chooses the same answer as the most influential incorrect human teammate, allowing it to “borrow” that teammate’s influence to sway the group, as well as to obfuscate its intentions.
>
> [1] P. Madhavan et al., Automation failures on tasks easily performed by operators undermine trust in automated aids, Human Factors, 2006

---

> ### Author Response · Authors · 2025-11-22
> **Part 2: Response to Weaknesses 3,4**
>
> **W3:** The reviewer’s suggestion regarding an additional “no information provided’’ baseline is reasonable. We do note that we intentionally opted not to test this baseline. Our primary objective was to demonstrate human vulnerability to trust-based attacks. To do that, the key comparison was between human behavior with a helpful AI versus a malicious AI. Moreover, given the degree of noise inherent in human studies, we believed it would be necessary to test each team under all three baselines in order to attribute differences between baselines to our experimental condition as opposed to team ability. This would have resulted in either very long experiments with too much cognitive load to the human participants or in very short sample trajectories where we could not meaningfully damage human behavior.
>
> **W4:** Yes, there are further experiments to verify the effectiveness of the cognitive model (see Fig. 6, Fig. 7, and Fig. 8 in Appendix B). We discovered post-hoc that the cognitive model substantially improves at predicting team score when the current performance features are included. This constitutes information that human participants themselves do not have, but which a malicious adversary would plausibly know. Unfortunately, our initial experiments were run using the cognitive model before we had developed the MLP, and our limited participant pool prevented rerunning all human-subject experiments with an improved cognitive baseline.
>
> This limitation is addressed in Lines 309-310 and Appendix A.2; however, we will expand our discussion to better explain (1) how the cognitive model improves under richer information, and (2) why its performance should not be interpreted as evidence that cognitive-based trust models are ineffective in general.

---

> > ### Comment · Reviewer_gYP2 · 2025-11-25
> > **Response**
> >
> > Thanks for the rebuttal which clarifies some concerns. Please include the necessary points into the revision. Particularly, I believe a straightforward takeaway would be beneficial, and a discussion on how AI researchers could benefit from the research would further strengthen the paper. Since I am not an expert in this AI deception domain, I prefer to maintain the score.

---

> ### Author Response · Authors · 2025-11-26
> **Thank you to reviewer gYP2**
>
> Thank you for your kind response to our rebuttal and for all your help in improving the quality of our paper! We have edited and uploaded a new version of our manuscript to reflect all of the points provided in our discussion. In particular, we have added a paragraph in our Discussion section titled “Summary of discussion” (Line 485) in which we provide a straightforward takeaway, as well as a discussion on how our research benefits inform both the real-world deployment of AI agents as well as how researchers could benefit from our contributions.
>
>
> For your convenience, we also include a copy of our added paragraph below:
>
> > Taken together, the paper’s results provide a key takeaway: even comparatively simple trust-based strategies can measurably harm the performance of human–AI teams, and this harm accumulates before humans recognize the manipulation. For real-world deployments, the paper highlights the need to anticipate adversarial behavior in AI assistants, rather than assuming benign operation. For research settings, the paper establishes a framework for studying trust-allocation and decision-making in human-AI teams, enabling systematic study of their robustness to trust-based attacks.

---

### Official Review · Reviewer_kEVp · 2025-11-01

**Soundness:** 2
**Presentation:** 2
**Contribution:** 2
**Rating:** 4
**Confidence:** 4

**Summary:**

In this paper, the authors propose an RL-based framework to control when an AI assistant should behave adversarially versus helpfully while collaborating with a group of three human participants in a sequential decision-making task. The AI attacker first behaves correctly to gain trust, and later selectively provides wrong answers to reduce the team’s overall performance based on the mdp planner. The authors later explore whether LLM agents can replicate the same patterns observed in humans under the attack scenario.

**Strengths:**

1. The problem of strategically harmful AI advice in human–AI teams is timely and relevant.
2. The task is small and interpretable, so it is easy to see how the attack works.
3. Comparing a psychology-based trust model and an end-to-end learned trust model  is  a reasonable way to study human influence modeling in the designed experiment.

**Weaknesses:**

1. Limited novelty over prior work [1]. The prior work (IJCAI'23) that already models human–AI interaction as an MDP and uses RL to strategically decide when the AI should behave adversarially to reduce human performance. This paper follows the same paradigm to decide when to deceive as action, and optimize long-term harm with rollouts using MDP. The main difference appears to be moving from a single human to a three-person team when interacting with the AI assistant, which is an incremental extension of the same paradigm.

2. Writing and problem framing are not coherent. Here are a few examples.
a) The introduction begins with a human–AI trust and safety story but ends by adding a new goal, namely checking whether LLM agents can replicate human behavior. This is not motivated by the earlier threat model and reads as disconnected from the main contribution.
b) The paper defines the reward strictly on the current round (the team score difference with and without the adversarial AI) but then claims to use dynamic programming and to “look ahead” several rounds. With a purely one-step reward there is no need to roll out future states, so either the actual objective is cumulative and not written, or the current formulation is incomplete. As written, the reward and the planner do not match.
c) The core method, UI, and evaluation are built for humans who explicitly allocate influence points. Dropping in LLM agents at the end does not follow from the setup and looks like content padding.

3. Experimental section is underspecified and weakly justified
a) Equal-weight baseline is not clearly defined. It is unclear whether it means uniform influence allocation, or simple averaging.
b) The psychology-based model performs poorly. In the comparison with the learned MLP, the model is close to or worse than the simple baseline, yet this is the model chosen for most of the RL rollouts because it is easy to simulate. This makes the justification of section 4.1 very weak as it does not fit the human behavior very well.

4. Unnatural human-study design
a) Unrealistic task-difficulty selection.
Participants choose the question difficulty themselves each round. It is unclear why a real-world collaborative AI setting would allow a team to self-select task difficulty. Most realistic environments assign tasks or randomize difficulty. Allowing participants to choose introduces potential confounding variables, and the paper does not explain what happens if participants disagree on difficulty.
b) Artificial “influence” mechanism.
The “influence” manipulation is not naturalistic. Participants see all answers and then must manually allocate a fixed “influence budget” to all four members every round. The team score is then mechanically computed from these allocations. This does not resemble real human-AI collaboration, where people discuss and form a joint decision or the system aggregates input automatically. The attack therefore exploits a contrived UI mechanic rather than demonstrating realistic persuasive or deceptive behavior. As a result, the validity and generalizability of the findings are limited to generalize well.











[1] Strategic adversarial attacks in ai-assisted decision making to reduce human trust and reliance. IJCAI 23

**Questions:**

see above weakness

---

> ### Author Response · Authors · 2025-11-22
> **Part 1: Response to Weaknesses 1,2**
>
> We thank the reviewer for their constructive assessment. Below we clarify our contributions, address issues of coherence, and explain design decisions in the modeling and human-subjects components. We also note that we have taken into account the reviewers feedback and have revised our manuscript accordingly.
>
> **W1:** We respectfully disagree that our contributions offer only incremental novelty over prior work [1]. The core objectives of the two papers are fundamentally different.
>
> In [1], the attacker is an external agent whose explicitly stated goal is “reducing human trust and reliance on the AI model”. The paper finds that the adversary reduces human trust more when it attacks the AI model on tasks with certain outcomes. Our goal is *instead* to design an adversarial AI so that it can mount an attack without being visible. In our setting, the attacker is the AI assistant, whose objective is to maximize long-term harm to team performance, explicitly balancing “harming team performance with loss of its own appraisal” (Lines 136–137). While the adversarial attack in [1] seeks to reduce human trust in AI, our goal is to maintain this trust while reducing the team’s performance. As a result of the difference in goals, in [1] the adversary attacks on the images in which the human is most confident, while in our case, the AI avoids disclosure by lying on the difficult questions in which the team of humans is the least confident. Besides the overall goal of why and how the adversarial attacks are being mounted, there are other differences in the two approaches such as individual versus team, classification score versus consensus score, 2-phase protocol versus one single phase, and the modeling of trust (as explained next).
>
> The modeling frameworks for trust differ in substance. Whereas [1] uses an MDP to predict a binary human adoption/rejection decision, we instead learn a row-stochastic influence matrix, capturing how influence is distributed across team members. This represents a substantially richer state representation and action space.
>
> Finally, extending from a single human-AI dyad to a three-person team is not a minor modification. It is well known in the literature that individual and group decision-making diverge due to social dynamics in team settings [2]. Structurally, the complexity of the system also changes: the dyad in [1] contains a single communication edge, while a team of three humans plus one AI produces twelve. Designing and experimentally validating an attack in this larger, socially coupled environment introduces significant new challenges that do not arise in the single-person case.
>
> [1] Z. Lu et al. Strategic adversarial attacks in ai-assisted decision making to reduce human trust and reliance. IJCAI, 2023
>
> [2] A. Ambrus et al. Group versus individual decision-making: Is there a shift?, RePEc, 2009
>
> **W2:** We appreciate the reviewer’s concern and will revise the introduction for clarity.
>
> (a) Our primary motivation for verifying whether LLM agents replicate human behavior is to determine if these agents can serve as human proxies in future studies. A key challenge in scaling human-subject studies is that human subjects are difficult to find and expensive to recruit. Therefore, by evaluating the behavior of LLM agents on our task, we believe we are contributing empirical evidence that helps answer a question of significant interest to our community.
>
> A second motivation comes from the rise of human-LLM teams, where a compromised LLM could amplify deception rather than merely fall victim to it. Prior work shows that LLMs can be attacked via prompt injection [1] and that humans are highly susceptible to LLM-generated misinformation [2]. This raises a key concern: if an LLM is vulnerable to deception, can an attacker exploit its vulnerabilities in ways that mislead both the model and its teammates? As we see in Fig. 5, LLM agents are indeed susceptible to the same trust-based attacks as humans, and CoT models do amplify deception. Our findings thus underscore the importance of further research in this area.
>
> (b) We will revise section 4.3 to clarify that our reward is cumulative. To clarify, although the per-round reward is defined on the immediate score differential, our planner evaluates the cumulative future reward in order to determine whether it should attack on a particular round. This is necessary to avoid losing significant trust early on, which may limit long-term ability to harm the team.
>
> (c) As a result of the reasons outlined in (a) and the belief that human-LLM systems will operate in shared environments, we intentionally test the performance of our LLM agents within the same framework as our humans. To address this concern, we clarify our motivation in Lines 90-91.
>
> [1] Z. Li et al. Evaluating the Instruction-Following Robustness of Large Language Models to Prompt Injection, EMNLP, 2024
>
> [2] C. Chen et al. Can LLM-Generated Misinformation Be Detected?, ICLR, 2024

---

> ### Author Response · Authors · 2025-11-22
> **Part 2: Response to Weaknesses 3,4**
>
> **W3:** (a) We will clarify in the manuscript that the equal-weights baseline assigns influence uniformly across all four agents.
>
> (b) While the psychology-based model underperforms the MLP at predicting full influence matrices, we highlight that both models achieve comparable accuracy at predicting team scores under the same information conditions. This is addressed in Lines 309 - 310 and shown in the Appendix: although influence-matrix errors differ measurably (Fig. 7), the two models track cumulative scores equivalently (Fig. 8). Because the RL attacker’s reward ultimately depends on score prediction, this provides the justification for using the psychology-based model in our rollouts.
>
> We also clarify that the psychology-based model was not selected because it was “easy to simulate.” Since our study focuses on human decision-making, we believed it was important to include a white‑box model grounded in the cognitive psychology literature (Lines 199-200). Moreover, training the MLP attacker required data from humans in our exact setting. Therefore, using the psychology-based model in early attacks allowed us to (a) evaluate a theoretically motivated model from the literature, (b) collect the human behavioral data needed to train the MLP, and (c) establish a baseline sense of human vulnerability to trust-based deception.
>
> Finally, the psychology-based model is not used in “most” of our RL rollouts. As stated on Line 375, it is used in 12 of 25 attacks, or approximately half. Human-subject constraints limited the total number of attack conditions we could test, so we implemented both attacker types in equal proportion to avoid over‑representing either condition.
>
> **W4:** (a) The intention was not to claim that real-world collaborative AI settings allows teams to self-select task difficulty. Rather, this mechanic instantiates a risk-reward trade-off, a central feature of real-world decision tasks. Teams can opt for safer low-value tasks or riskier high-value ones, which allows us to observe how adversarial AI affects group reasoning under uncertainty. We will clarify this rationale and note that disagreements within teams are resolved through discussion before selecting a difficulty (Lines 170-172).
>
> (b) We agree that the influence allocation interface is artificial, but emphasize that this was the end result of an iterative process. In early pilots, we attempted a more naturalistic design where teams discussed their answers and reported influence through post-round surveys. Unfortunately, participants found these surveys burdensome and responses quickly became noisy or random. We then considered the possibility of sampling influence less frequently; however, this would even further shrink an already small dataset of 625 datapoints. Finally, we realized that participants were deeply engaged with the game and highly motivated to score highly. As a result, by coupling the influence measurement with the scoring mechanism, we were able to leverage the score incentive to reliably collect high-quality data on influence allocation. We will clarify this limitation in Section 8 and note that studying fully naturalistic influence evolution is important future work but beyond the scope of this study. Lastly, while the influence interface may limit the realism of our specific task (a trivia-style game), it does not undermine the broader validity of the framework itself: deceptive attacks that exploit trust evolution remain applicable beyond the interface used to measure influence.

---

### Official Review · Reviewer_DHYo · 2025-11-01

**Soundness:** 3
**Presentation:** 3
**Contribution:** 3
**Rating:** 4
**Confidence:** 3

**Summary:**

This paper studies how adversarial AI agents can manipulate human-AI teams using Model-Based Reinforcement Learning. In a trivia game setup, the AI learns to exploit human trust dynamics, reducing team performance. A cognitive and a data-driven MLP method are compared, with the MLP model proving more effective. The study also shows that LLM-based teams display human-like trust behaviors and are similarly vulnerable to attacks.

**Strengths:**

- The paper is proposing an interesting approach to attack humans and AI collaboration by modeling humans' trust evolution.
- The proposed method is both evaluated with a human in the loop and with LLM. The evaluation shows the proposed attack is able to reduce the overall team performance.

**Weaknesses:**

- This paper only evaluated using only 1 type of game. It is unclear how and if the proposed method can be generalized to more complex scenarios where it is non-trivial to get the specific intermediate metric for
- This work seems to be missing discussions of related works on attacks in multi-agent reinforcement learning literature, which can have a similar setting to this paper.
- There seems to be limited performance comparison to other baseline attack methods, such as the ones that focus on attacking cooperative MARL or even a random selection policy for the AI attacker.

**Questions:**

- What are the specs of the inputs to the MLP? The paper mentions that "round number, the current performance (c.p.) of the human and AI agents, and a summary of past correct answers" are used as input. How are these encoded and sent to the MLP?
- It seems the MLP is not using language/text information. Would it have higher performance if you also include text information there? The authors claim that the model based on principles of cognitive psychology does not lag too far behind the MLP model, does this still hold if you use more advanced method?
- For the evaluation, how much of the collected human data is used for training and how much is used for evaluation for the accuracy of the human model?
- Do these same approaches work for other games, especially the ones with high variance in the team score?

---

> ### Author Response · Authors · 2025-11-21
> **Part 1: Response to Weaknesses**
>
> We thank the reviewer for the thoughtful and thorough review of our work. We appreciate the constructive feedback, and we are glad that the reviewer recognized the novelty and contributions of our approach. We also note that all modifications discussed in our response are reflected in our revised manuscript.
>
> **W1:** The main contribution of our work is to experimentally demonstrate that even in a simple setting, strategically optimized deception can produce measurable and statistically significant harm to real human teams. Although our approach of using influence scores is tailored to our domain, we believe our framework of modeling trust evolution and strategically attacking is domain agnostic. Therefore, the key challenge for generalization to more complex scenarios would be in determining and modeling alternative proxies for trust evolution as well as devising domain-specific attackers. We believe that both of these areas remain an open research direction outside of the scope of this initial study, and we clarify these limitations and future directions in our Sec 8 of our manuscript under “Influence allocation mechanism."
>
> **W2:** We agree that our manuscript could benefit from a discussion of related works in MARL literature. To acknowledge this, we have drafted a literature review paragraph focused on the connection between our work and MARL. First, we introduce [1,2] to highlight that cMARL strategies are susceptible to adversarial attacks. Then, from [3], we see that it is possible to design RL agents who are capable of disguising their reward function (and therefore intent) from humans. This opens up the possibility for RL agents to maliciously attack teams unnoticed. Finally, [4] practically demonstrates that a single black-box RL attacker can leverage the influence between agents and their cooperative nature to learn a policy to harm the team. This paragraph positions our work as an extension of this space by demonstrating a practical RL-based attack on a human team rather than a team of artificial agents.
>
> [1] Y. Huang et al. Deceptive Reinforcement Learning Under Adversarial Manipulations on Cost Signals, 10th International Conference on Decision and Game Theory for Security, 2019
>
> [2] Y. Hu et al. Sparse adversarial attack in multi-agent reinforcement learning, arXiv, 2022. URL https://arxiv.org/abs/2205.09362.
>
> [3] Z. Liu et al. Deceptive Reinforcement Learning for Privacy-Preserving Planning, AAMAS, 2021
>
> [4] S. Li et al. Attacking Cooperative Multi-Agent Reinforcement Learning by Adversarial Minority Influence, Neural Networks, 2025
>
> **W3:** Although incredibly exciting theoretical, computational, and practical work has occurred in the study of MARL, to the best of our knowledge, this study is the first that demonstrates RL-based attacks with real mixed human-AI teams in the loop.
>
> Unfortunately, in this type of human-subject study, implementing attack strategies from the cMARL literature is infeasible due to practical limitations on the duration of our experiment and the availability of data. Our attack would fall under the paradigm of policy interference attacks. We note comparable strategies (e.g., [1]) collect as many as 100x10^6 data points. This is true for other strategies too. For example [2] trained an agent for 10,000 episodes of 50 steps to mount a communication attack. Although our action space is small, it is clear that collecting equivalent data from human-subjects to train such strategies is not realistic.
>
> For the same reasons above, we opted not to conduct a random-selection baseline. We believe this baseline would not be exceptionally informative relative to the no-attack baseline where the AI agent has accuracy of 75% (vs. random 50%). Anecdotally, we observed rapid loss of trust in the AI if it was observed to regularly make mistakes on easy questions. Therefore, we also expect a random lie policy would either: (a) drop accuracy unpredictably, or (b) collapse trust immediately.
>
> [1] S. Li et al. Attacking Cooperative Multi-Agent Reinforcement Learning by Adversarial Minority Influence, Neural Networks, 2025
>
> [2] Y. Sun et al. Certifiably Robust Policy Learning against Adversarial Communication in Multi-agent Systems, ICLR, 2023

---

> ### Author Response · Authors · 2025-11-21
> **Part 2: Response to questions**
>
> **Q1:** The model architecture consists of an input layer, three hidden layers, and an output layer. Our original model (used for our MLP attack experiments) is trained solely from our initial 10 cognitive attack teams (i.e., 250 data points). This model uses a batch size of 128, a learning rate of 0.01, and 100 epochs with early stopping. Our final model utilizes more data and is instead trained with a batch size of 64, a learning rate of 0.001, and for 350 epochs with early stopping. For input, our chosen features are organized into a vector and passed to the model. The model outputs a vector of dimensionality 12 which we reshape into a 3x4 matrix. After computing the row-wise soft-max and re-normalizing, we obtain our row-stochastic influence matrix. We will add a description of our network in our methodology.
>
> **Q2:** We experimented with simple text-derived features such as the number of messages sent by each player, but these did not improve model performance. In fact, our experiments found a slight deterioration in performance. More specifically, compared to Fig. 2 from the paper, our cumulative error in score prediction for the MLP model increased by two points. The MSE for the influence matrix prediction also negligibly increased to approximately 0.10. We do note that prior work has found that incorporating richer text information leads to improvements in predictive power, and we hope to eventually integrate LLM reasoning into our ML pipeline to evaluate whether such hybrid models significantly alter deceiver performance in our context.
>
> **Q3:** We train on 19 teams and evaluate on the remaining 6, choosing this split to maintain a balanced test set (3 MLP-attacked teams, 3 cognitive-attacked teams). In other words, we have 475 data points for training and 150 for testing.
>
> **Q4:** Assuming that a “high-variance” environment is one with significant noise, we believe our attack would still work. However, we note that the performance of our attack will most likely decrease proportionally to the signal-to-noise ratio. Intuitively, the fundamental signal in our attack is the degree of misplaced trust. If we can not effectively measure that signal (relative to well-placed trust), then we can not decide when to attack.

---

### Official Review · Reviewer_TNzR · 2025-11-02

**Soundness:** 3
**Presentation:** 3
**Contribution:** 4
**Rating:** 8
**Confidence:** 3

**Summary:**

This paper explores how an adversarial AI assistant can intentionally harm decision-making in human-AI teams using a model-based reinforcement learning (MBRL) framework. In a trivia-style game with 25 teams of humans, the AI learns to model human trust evolution and strategically misleads them to degrade performance. Two models, one grounded in cognitive psychology and another data-driven (MLP), serve as the attacker’s internal models. The results show that both attacks successfully reduce team performance, with the data-driven model being more effective. The authors further simulate similar attacks on teams of large language models (LLMs), finding that even advanced models show human-like vulnerabilities to manipulation.

**Strengths:**

The paper tackles an important and timely topic, studying trust manipulation in human-AI collaboration, through a creative experimental setup.

It integrates psychological theory with reinforcement learning in a clever way, grounding the attack models in cognitive principles.

The experimental design (human-AI trivia teams) is easy to understand and allows for clear quantification of performance degradation.

The inclusion of both human and LLM “team” experiments broadens the relevance of findings to real-world AI safety concerns.

The analysis connects observed behaviors, such as over-reliance on AI or trust collapse, to known human cognitive biases, adding interpretability to technical results.

**Weaknesses:**

The human study is well executed and valuable given how hard it is to recruit real participants for this kind of setup, but since it focuses only on a short trivia-style game, it’s still unclear how the findings would generalize to other types of collaborative tasks, higher-stakes environments, or longer-term interactions where trust and deception evolve differently.

The setup feels a bit too clean and simplified; it’s a short trivia game where the AI either lies or tells the truth, which doesn’t fully capture how messy and nuanced deception or teamwork can be in real life.

The models assume people update trust in a logical, almost math-like way, but real human trust involves a lot more emotion, tone, and social context, and the validation study focuses more on plausibility than real persuasion or manipulation effects.

The LLM simulation part isn’t explained clearly enough; it’s not totally clear whether these models actually mimic human trust patterns or if they’re just picking up on easy cues from the chat history.

The ethical side feels a bit light; there’s little mention of how participants were debriefed after being deceived, and releasing attack methods like this could backfire without stronger guardrails or usage guidelines.

**Questions:**

How exactly does the RL attacker balance “lying frequency” with maintaining trust? Does the model explicitly optimize for long-term believability?

Were participants aware that the AI might perform poorly or inconsistently, or was full deception used during the study?

How were the trivia questions selected and verified to avoid bias toward topics the AI might be stronger in?

How were LLM “teams” operationalized? Were they role-playing humans, or simply optimizing point distribution based on chat logs?

If the attacker learns human trust evolution, what prevents it from exploiting individual differences (e.g., cautious vs. trusting participants)?

**Details Of Ethics Concerns:**

No major ethical violations are apparent since IRB approval was obtained and participant deception was minimal. However, a short internal ethics check focused on AI manipulation and dual-use risk could still be advisable.

---

> ### Author Response · Authors · 2025-11-21
> **Part 1: Response to Weaknesses (and Q2)**
>
> We thank the reviewer for their thoughtful and detailed feedback and hope they find our response below to each Weakness (W1-5) and Question (Q1-5) helpful. These points have also been addressed in our main manuscript.
>
> **W1:** We acknowledge that our setup does not capture more complex collaborative interactions. This design was chosen in consideration of our human subjects. In early iterations, we experimented with multiple trivia categories, additional questionnaires and surveys on influence/confidence, and longer game horizons (up to two hours). Even with these longer sessions, participants could not meaningfully explore a larger action space, and the cognitive load led to significant social loafing and unusable survey responses. To maintain engagement while still observing meaningful human-AI interactions, we reduced both the length of the experiment and the action space. Despite this constraint, our results demonstrate that AI can exploit human trust patterns even in a highly controlled setting, and we leave broader generalization to other tasks and environments as future work.
>
> **W2 + W3:** We agree that our simple task does not capture the full emotional and social complexity of real-life human teams; however, this simplicity is intentional. Our study applies the methodological approach common in experimental sociology to understanding decision-making in human-AI teams. As part of this framework, we attempt to isolate the variable of interest (i.e., trust evolution) from confounders such as human biases and emotions. We also focus on a trivia game with objective answers to create a setting where it is reasonable for trust to evolve logically, allowing us to draw conclusions on the capacity of a malicious agent to inflict harm. We view our research as complementary to other studies which may examine trust evolution in richer environments.
>
> Finally, we believe our results go beyond demonstrating plausibility given that we also demonstrate real effects of our manipulation. For example, Fig. 3 reveals individual trends in influence allocation before and after the attack, and Fig. 4 illustrates statistically significant reductions in team performance. We clarify this point by revising the corresponding sections (Sec 5.2, Sec 5.3, Fig. 4 caption) and discussion (under “Efficacy of attacks on Human-AI Teams”).
>
> **W4:** We believe our LLM agents both mimic human trust patterns and pick up on easy cues from the chat history. As shown in Figure 5, without chat history, the cumulative error relative to humans in the LLM score trajectory is approximately three points, or a single round of value. This indicates the model is already capable of mimicking human trust patterns. By including the chat, the model is able to also pick up on easy cues from the chat history to refine its allocation and further close the gap with the average human behavior.
>
> **W5 + Q2:** We appreciate your concern for ethical research and share in your commitment. Our consent form explicitly states that “the accuracy of the answer [provided by the AI] is not guaranteed” and that “it will be up to your group to decide whether to use the AI, when to use it, and if the answer can be trusted.” All participants were fully debriefed at the end of the study, including an explanation of the AI’s deceptive behavior and the purpose of the design.
>
> Moreover, while any research involving deception carries inherent risk, we believe understanding these vulnerabilities offers important societal value. Our attack is intentionally weak (MLP + DP) compared to modern systems that already display more sophisticated deceptive tendencies (e.g., LLM sycophancy). We therefore do not anticipate our work being used to cause widespread harm.
>
> We clarify all of these points within our revised Ethics section.

---

> ### Author Response · Authors · 2025-11-21
> **Part 2: Response to Questions**
>
> **Q1:** The model implicitly optimizes for long-term believability as a natural consequence of the reward design. As we see in Eqs. (3) and (5), the attacker is rewarded for minimizing the expected team score under attack. Achieving this requires the team to allocate influence to the AI (see Eqs. (4) and (5)), and therefore the agent must avoid actions that lose too much trust.
>
> **Q3:** We assume a hypothetical white-box attacker and focus on assessing attack vulnerability. In other words, the AI does not rely on the content of the trivia questions. As described in Sec. 4.4, for the first 10 rounds the AI responds with a fixed accuracy, and after round 10 it only decides when and how to lie.
>
> **Q4:** The LLMs were told that they were members of the team responsible for allocating influence points. They were not role-playing humans. In addition to chat logs, they were also given round-by-round data as described in Sec. 4.5. The specific prompt can be found in Appendix E.
>
> **Q5:** Nothing in our framework explicitly prevents the attacker from exploiting individual differences. However, because the reward is defined on a group level, focusing on a single player may increase the likelihood of teammates noticing the inconsistency and reducing their trust in the AI, thus reducing the attacker’s expected reward.

---

### Author Response · Authors · 2025-12-02
**Summary of Discussion and Changes for AC**

Dear Area Chair,

Below, we provide a consolidated summary of our revisions and responses to all reviewers. We have worked diligently to update our manuscript to address every concern of our paper and we are thankful to our reviewers for their thoughtful comments and help improving our manuscript.

---
# Summary of Reviews
Overall, our reviewers agreed that our paper addresses a timely and important problem in human-AI collaboration, namely the vulnerability of human agents to strategic manipulation. Reviewers valued our connections to cognitive psychology, believing it made our results more interpretable, while also noting the relevancy of our findings to real-world human-AI safety.

By addressing all reviewer feedback, our revisions substantially improve the quality of our paper. Specifically, our revised manuscript:
1. **Enhances technical clarity** by adding a summary of contributions to our introduction, a summary of our takeaways to our discussion, and elaborating on technical details in our methods, results, and appendices
2. **Addresses conceptual novelty and positioning** by adding a discussion of cMARL in our Related Works section and clearly articulating how our multi-human, trust-preserving setting differs significantly from prior work, specifically an IJCAI 2023 paper.
3. **Motivates the need to evaluate LLMs on our task** by emphasizing the connection between our results to the emergent paradigm of human-LLM teams, as well as addressing the growing interest for LLMs as human proxies.
4. **Addresses ethical concerns of attacking humans** by revising our Ethics section to elaborate on how our human subjects were informed of the AI's nature, debriefed, and the value of such experiments.

---
# Detailed Responses to All Reviewers

### **Reviewer TNzR - Generalizability, task realism, ethics of attack**

**Original Score:** 8

**Our Response:**

- **Addressed concerns about generalizability** by clarifying that trust-based attacks are not limited to our specific environment and that our setting serves as a controlled proof-of-feasibility.
- **Justified the experimental design** by elaborating on limitations observed in our pilot studies and how those constraints shaped the final configuration.
- **Motivated minimalist environment** as a deliberate choice to isolate trust dynamic
- **Clarified details** regarding LLM setup and behavior on task
- **Substantially expanded Ethics section** to address concerns regarding consent and debriefing
---
### **Reviewer DHYo - Generalizability and connections to MARL**

**Original Score:** 4

**Our Response:**
- **Revised related works section** to situate our work within cMARL and adversarial MARL literature
- **Provided justification** for why MARL-style baselines are infeasible in this domain and clarified our resulting baseline choices.
- **Revised appendix** to include complete information regarding model architectures and training
- **Justified generalizability of trust-based attacks** despite our specific implementation being limited to our domain
---
### **Reviewer kEVp - Technical clarity, novelty over prior work, task realism**

**Original Score:** 4

**Our Response:**
- **Justified novelty over prior work** by noting explicit differences between objectives as well as explaining the significance of multiple agents to task complexity
- **Added contribution statement to introduction** to explicitly state outcomes
- **Revised introduction** to improve coherence and justify inclusion of LLMs on task
- **Revised methods section** to provide additional clarification on cumulative reward structure and equal-weights baseline
- **Revised experimental setup section** to explain our motivation for allowing teams to self-select difficulty level
- **Justified the experimental design** by explaining that the final configuration resulted from an iterative process aimed at isolating and measuring trust evolution.
---
### **Reviewer gYP2 - Clarity of technical contributions and takeaways**

**Original Score:** 6

**Our Response:**
- **Revised discussion section** to provide straightforward takeaway and explain how AI researchers could benefit from our work
- **Added summary of contributions to introduction** to improve clarity
- **Justified chosen baselines** and explained why we did not use a “no information” baseline
- **Addressed concerns regarding efficacy of cognitive model** by referencing additional experiments in our appendices

---
# Conclusion

With our revisions, we believe our paper represents a strong and novel contribution to the field of multi-agent decision making with important implications for our primary areas of alignment, safety, and societal considerations.

We hope that you find our work interesting and kindly request your favorable consideration for acceptance to ICLR 2026.

Sincerely,

The Authors

---

### Meta-Review · Area_Chair_U8hj · 2026-01-08

**Summary:**

This paper studies manipulation of human-AI teams by adversarial AI agents, leveraging methods from model based RL and using a trivia game as a testbed.

The concerns brought up by the reviewers were about novelty of the work, positioning w.r.t. related work, and limited empirical evaluations (DHYo: "This paper only evaluated using only 1 type of game."; TNzR: "The setup feels a bit too clean and simplified; it’s a short trivia game where the AI either lies or tells the truth, which doesn’t fully capture how messy and nuanced deception or teamwork can be in real life."). The authors addressed most of the concerns related to novelty, ethical concerns, and positioning of the work; the critique of the simplicity of chosen setup was addressed by emphasizing that such simplicity is their design choice and allowed the authors to isolate trust dynamics.

The fact that conclusions may not generalize to other even simple games remained an outstanding issue (i.e., it would've been more convincing if the authors could have conducted their study on multiple games to demonstrate that conclusions remained the same). However, the paper's value lies in providing a solid framework and starting point for future research to explore these questions. Additionally, the extensive human study is a significant contribution that is not common.

Since the authors have carefully addressed all other concerns and the reviewers did not identify any fatal flaws, my recommendation is that the paper's potential for impact warrants an accept.

**Reviewer Concerns:**

- The authors nicely summarized all reviewer concerns and their response to these concerns in https://openreview.net/forum?id=Lqt5weP0Gr&noteId=Npo8BrtGBK
- All concerns except limitations of the experimental setup and the fact that conclusions may not generalize have been addressed satisfactorily. Concerns about generalization to more settings is a good avenue for future research by the authors and/or community.

**Reviewer Scores:**

My best assessment based on the discussion is that all reviewers would have likely maintained or slightly increased their scores.

---

### Decision · Program_Chairs · 2026-01-26

Accept (Poster)